# Role of Running-Activated Neural Stem Cells in the Anatomical and Functional Recovery after Traumatic Brain Injury in p21 Knock-Out Mice

**DOI:** 10.3390/ijms24032911

**Published:** 2023-02-02

**Authors:** Jonathan Isacco Battistini, Valentina Mastrorilli, Vittoria Nicolis di Robilant, Daniele Saraulli, Sara Marinelli, Stefano Farioli Vecchioli

**Affiliations:** 1Institute of Biochemistry and Cell Biology, Institute of Biochemistry and Cell Biology, National Research Council (IBBC/CNR), Monterotondo, 00015 Rome, Italy; 2Plaisant S.R.L., 00128 Rome, Italy; 3Experimental Translational Oncology Department at Menarini Ricerche, Pomezia, 00071 Rome, Italy; 4Department of Law, Economics, Politics and Modern Languages, LUMSA University, 00193 Rome, Italy

**Keywords:** subventricular zone, adult neurogenesis, neural stem cells, traumatic brain injury, p21

## Abstract

Traumatic brain injury (TBI) represents one of the most common worldwide causes of death and disability. Clinical and animal model studies have evidenced that TBI is characterized by the loss of both gray and white matter, resulting in brain atrophy and in a decrease in neurological function. Nowadays, no effective treatments to counteract TBI-induced neurological damage are available. Due to its complex and multifactorial pathophysiology (neuro-inflammation, cytotoxicity and astroglial scar formation), cell regeneration and survival in injured brain areas are strongly hampered. Recently, it has been proposed that adult neurogenesis may represent a new approach to counteract the post-traumatic neurodegeneration. In our laboratory, we have recently shown that physical exercise induces the long-lasting enhancement of subventricular (SVZ) adult neurogenesis in a p21 (negative regulator of neural progenitor proliferation)-null mice model, with a concomitant improvement of olfactory behavioral paradigms that are strictly dependent on SVZ neurogenesis. On the basis of this evidence, we have investigated the effect of running on SVZ neurogenesis and neurorepair processes in p21 knock-out mice that were subject to TBI at the end of a 12-day session of running. Our data indicate that runner p21 ko mice show an improvement in numerous post-trauma neuro-regenerative processes, including the following: (i) an increase in neuroblasts in the SVZ; (ii) an increase in the migration stream of new neurons from the SVZ to the damaged cortical region; (iii) an enhancement of new differentiating neurons in the peri-lesioned area; (iv) an improvement in functional recovery at various times following TBI. All together, these results suggest that a running-dependent increase in subventricular neural stem cells could represent a promising tool to improve the endogenous neuro-regenerative responses following brain trauma.

## 1. Introduction

Traumatic brain injury (TBI) is one of the most common causes of death and disability in young people [1]. TBI severity is assessed on the basis of the Glasgow Coma Scale (GCS) score [2], with further modifications and can be graded as mild, moderate or severe [3,4]. The symptoms of mild patients are short-term memory and concentration difficulties and they usually show complete neurological recovery [5]. Moderate patients are lethargic and stuporous, whilst severe subjects are comatose, unable to open their eyes or follow commands [6]. Severe patients also have a higher risk of hypotension, hypoxemia and brain swelling, all effects that, if not prevented, can cause complications and lead to death [7,8]. Moreover, TBI strongly increases a patient’s susceptibility to neurodegenerative diseases such as Alzheimer’s and Parkinson’s disease [9,10].

The adult mammalian brain is able to respond to damage with structural and functional modifications and regeneration mechanisms. In this context, the role of the endogenous neurogenic post-traumatic response is the subject of a wide range of studies [11,12]. Under physiological conditions, the neural stem cells (NSCs, named type B cells) that reside in the SVZ represent a population of relatively quiescent cells [13,14], which give rise, in the course of neurogenic differentiation, to the following two classes of cells: type C cells that have a high proliferative rate [15], which in turn give rise to neuroblasts called type A cells [16]. These cells exit the cell cycle and migrate along the rostral migratory stream to reach the olfactory bulb, where they mature into inhibitory GABAergic neurons [17,18]. After brain injury, endogenous neural stem cells can be activated to modulate the timing and rate of proliferation/differentiation, with the aim of facilitating brain repair [19,20]. This process is also finely regulated by a series of changes in the environment of the neurogenic niche, such as an increase in vasculature permeability that favors the migration of NSCs and neuroblasts to the injured cortex [21]. Furthermore, transcriptomic studies based on single-cell RNA sequencing (RNA-seq) have shown a strong increase in the post-injured SVZ of the proportion of “primed” quiescent and active NSCs, which highly express genes related to protein synthesis and cell cycle regulation [22]. These findings highlighted the existence of reactive SVZ NSCs capable of conferring protection following TBI, suggesting an important role of the activation of SVZ NSCs in providing beneficial outcomes for post-TBI brain repair.

The p21^Waf1/Cip1^ gene represents one of the main regulators of the cell cycle and plays a primary role in modulating the transition between quiescence and activation of NSCs in adult neurogenic niches [23]. This gene is part of the Cip/Kip family of cyclin-dependent kinase inhibitors (CKIs), which also include the p27 and p57 genes with the function of negatively regulating cell cycle progression [24,25]. Within neurogenic niches, p21 expression correlates with the maintenance of NSCs in a quiescent state and with the restraining of progenitor proliferation [26,27,28]. Constitutive deletion of the p21 gene in mouse models has led to the rapid and powerful activation in the cell cycle of quiescent NSCs at the post-natal stage, with a consequent reduction in the self-renewal capacity of NSCs, the onset of replicative stress in the hyper-proliferating NSCs and progenitors and a reduction in neurogenesis in adult mice [28,29,30,31]. Previous work by our group demonstrated that in the absence of the p21 gene, SVZ neurogenesis and olfactory behavior are significantly enhanced by 12 days of voluntary running. These results strongly indicate that p21-null NSCs retain their high neurogenic potential, which is specifically triggered by physical activity [28]. Based on these findings, in this project, we investigated the possibility that the enhancement of SVZ neurogenesis that occurs in a p21-null mouse model that undergoes a 12 day-session of voluntary running could be an effective mechanism that contributes to the neuroanatomical and functional recovery processes following pathological conditions such as TBI. Our results demonstrate that at different time-points following TBI, the combination of running and of p21 knockdown induces an increase in the migration of SVZ NSCs and progenitors toward the cortical lesion, an enhancement of new differentiating neurons in the peri-lesioned area and partial functional recovery in the injured mice. These data suggest a potential model for the strengthening of post-traumatic endogenous neurogenesis that is capable of effectively counteracting the anatomical and functional dysfunctions induced by cortical damage and accelerating neurorepair processes.

## 2. Results

The aim of this work is to analyze the post-traumatic neurogenic effects of a running session of 12 days in a mouse model with the deletion of p21. The 12-day running paradigm was chosen in accordance with previous data that demonstrate that this running protocol provokes the peak of running-dependent increments of neurogenesis in the SVZ of p21 ko mice [28]. The experimental procedure is detailed in the Material and Method section and is shown in Figure 1.

### 2.1. TBI Induces the Significant Activation of Type B NSCs in the SVZ of Injured Mice

To understand the impact of TBI on type B Glia-like NSC recruitment and proliferation, we analyze, at different time points from the trauma (7, 14 and 30 days), the ipsi- and contralateral SVZ of mice subjected to brain injury or in the SHAM condition. The NSC sub-population was identified through the co-localization of the GFAP marker and the Nestin GFP transgene.

The data show a strong increase in the SVZ of both hemispheres of type B cell recruitment from quiescence in the TBI groups, in comparison with their SHAM counterpart at 7 days post TBI (recruitment: ratio of Ki67^+^NestinGFP^+^GFAP^+^ cells/NestinGFP^+^GFAP^+^ total cells, ipsilateral: *p* < 0.001, Figure 2A–C, Appendix A; contralateral: *p* < 0.001, Appendix A) and their proliferation (Ki67^+^NestinGFP^+^GFAP^+^ cells, ipsilateral: *p* < 0.001, Figure 2A,B,D; contralateral: *p* < 0.001, Appendix A), as well as 14 days after injury (recruitment: ipsilateral, *p* < 0.001, Figure 2E, Appendix A; contralateral: *p* < 0.001, Appendix A; proliferation: ipsilateral: *p* < 0.001, Figure 2F, Appendix A; contralateral: *p* < 0.001, Appendix A).

After 30 days from TBI, we observe in the SVZ of both hemispheres a main lesion effect of the type B cells’ pool of TBI groups compared to the SHAM animals, due to the strong increase observed in the KO TBI and KO RUN TBI mice, compared to the respective SHAM mice (NestinGFP^+^GFAP^+^ cells, ipsilateral: *p* < 0.001, Figure 2G, Appendix A; contra-lateral: *p* = 0.002, Appendix A).

These data let us hypothesize that TBI might induce the powerful activation of type B cells, in terms of recruitment from quiescence and proliferation, leading in the long term to an expansion of the pool compared to the SHAM groups.

### 2.2. Influence of p21 Deletion and Running Session on TBI-Induced NSC Activation

Moreover, we evaluated the different responses to trauma within the injured groups, with the aim of assessing whether the deletion of p21 and/or the physical activity before the TBI might promote the neurogenic response. The comparative analysis within the four TBI groups (WT TBI, WT RUN TBI, KO TBI and KO RUN TBI) indicates that at 7 days post trauma, no changes were detectable in the recruitment or expansion of type B cells. Instead, after 14 days, we observe, in the contralateral SVZ, a TBI-dependent expansion of type B cells in the KO RUN TBI, with respect to the other groups in terms of the enhancement of type B proliferation (KO RUN TBI vs. WT TBI and KO TBI, *p* < 0.001, Appendix A). In addition, 30 days after the TBI, in the ipsilateral SVZ, no significant variations were detected, while in the contralateral SVZ, a strong expansion of type B pool size in the KO RUN TBI mice was detected (NestinGFP^+^GFAP^+^ cells: KO RUN TBI vs. WT TBI, WT RUN TBI and KO TBI, *p* < 0.001, Appendix A).

From this first analysis, it emerges that the deletion of p21 and/or physical activity is ineffective in triggering the post-traumatic NSC response in the ipsilateral hemisphere, while a strong effect of running and p21 deletion on NSC activation is observed in the contralateral hemisphere.

### 2.3. Time-Course of Neural Stem Progenitor Cell (NSPC) Modulation after TBI

The count of NestinGFP^+^ cells and their proliferating fraction (Ki67^+^ NestinGFP^+^ cells) allowed us to evaluate the variations in the stem cells and the transit amplifying progenitors (small fraction of type B and C cells), hereinafter collectively referred to as neural stem/progenitor cells (NSPCs).

At 7 days post TBI, we observe in both ipsi- and contralateral regions a strong increase in NSPC proliferation in the WT and KO groups subjected to TBI, with respect to their SHAM groups (Ki67^+^NestinGFP^+^ cells, ipsilateral: WT TBI vs. WT SHAM and KO TBI vs. KO SHAM *p* = 0.015, Figure 3A,C, Appendix A; contralateral: *p* = 0.013, Appendix A), which induces a significant increase in total proliferation (Ki67^+^ cells, ipsilateral WT TBI vs. WT SHAM, *p* = 0.03, KO TBI vs. KO SHAM *p* < 0.001, Figure 3A,D; contralateral: *p* = 0.0013, Appendix A). On the other hand, in the KO RUN TBI group, we detect in the ipsi- and contralateral SVZ a net decrease, compared to the KO RUN SHAM group, of proliferating NSPCs (Ki67^+^NestinGFP^+^ cells, ipsilateral: KO RUN TBI vs. KO RUN SHAM, *p* = 0.009, Figure 3A,C, Appendix A; contralateral: KO RUN TBI vs. KO RUN SHAM *p* < 0.001, Appendix A) as well as of total proliferation (Ki67^+^ cells, ipsilateral: KO RUN TBI vs. KO RUN SHAM, *p* < 0.001, Figure 3A,D, Appendix A; contralateral: KO RUN TBI vs. KO RUN SHAM *p* < 0.001, Appendix A).

The analysis carried out 14 days post TBI demonstrates a significant decrease in the SVZ of both hemispheres of the NestinGFP^+^ cell pool in all the TBI groups compared to their SHAM counterparts (ipsilateral: *p* < 0.001, Figure 3B,E; contralateral: *p* = 0.0011, Appendix A). Moreover, we observe in the ipsilateral hemisphere of the WT TBI and KO RUN TBI groups a sharp decrease in total proliferation, compared to the respective WT SHAM and KO RUN SHAM groups (Ki67^+^ cells: WT TBI vs. WT SHAM, *p* = 0.029; KO RUN TBI vs. KO RUN SHAM *p* < 0.001, Figure 3B,F), as well as in the number of Ki67^+^ NestinGFP^+^ cells (WT TBI vs. WT SHAM, *p* = 0.06; KO RUN TBI vs. KO RUN SHAM *p* = 0.04).

After 30 days from the TBI, we observed the main lesion effect on the decrease in NestinGFP^+^ cells in the ipsilateral SVZ of the TBI groups compared to the SHAM mice (*p* = 0.003, Figure 3G).

These data demonstrate the initial expansion of NestinGFP^+^ cells in the WT TBI and KO TBI mice, while a decrease in NSPCs is observed in the KO RUN TBI group with respect to the KO RUN SHAM mice, which display a powerful increase in SVZ neurogenesis, as previously shown [28]. Later, the decline in NestinGFP^+^ cells also becomes evident in the other TBI groups compared to their SHAM counterparts.

### 2.4. Influence of p21 Deletion and Running Session on NSPC Regulation after TBI

The analysis within the injured groups did not reveal at 7 days post TBI any significant differences in the NestinGFP^+^ sub-populations. In addition, 14 days after TBI, we observe in the ipsilateral SVZ an increase in the proliferating Nestin GFP^+^ and in the total proliferation in the WT RUN TBI, KO TBI and KO RUN TBI group compared to the WT TBI group (WT TBI vs. WT RUN TBI, *p* = 0.01, vs. KO TBI and KO RUN TBI *p* < 0.001; Ki67^+^ cells: WT TBI vs. WT RUN TBI, *p* = 0.01, vs. KO TBI, and KO RUN TBI, *p* < 0.001, Figure 3F). In the contralateral SVZ, we observe a strong proliferative response of NestinGFP^+^ cells in the KO RUN TBI group, compared to the WT TBI mice (Ki67^+^ NestinGFP^+^: KO RUN TBI vs. WT TBI, *p* = 0.024, Appendix A), leading to an increase in NestinGFP^+^ cell pool size (KO RUN TBI vs. KO TBI, *p* = 0.002, Appendix A), and consequently in total proliferation (KO RUN TBI vs. WT TBI, *p* = 0.0048, vs. KO TBI, *p* = 0.006, Appendix A). At 30 days post TBI, we did not find any difference within the injured groups in either of the hemispheres.

These data suggest a transitory increase 14 days after the TBI in contralateral subventricular cell proliferation, which is dependent on physical activity and the lack of the p21 gene.

### 2.5. TBI Triggers a Late Increase in Type A Neuroblasts

The analysis of the TBI-induced modulation of type A neuroblasts was carried out by using the specific marker DCX. At 7 and 14 days post TBI in the ipsi-lateral region, we did not detect any significant variation in DCX^+^ neuroblasts in the injured groups compared to the corresponding SHAM groups, while in the contralateral SVZ, our data showed at 7 days post TBI a lesion effect on the increase in neuroblasts of the injured groups, compared to their SHAM counterparts (DCX^+^ cells: *p* = 0.004, Appendix A); after 14 days post TBI, we observed in the contralateral SVZ a lesion effect on the decrease in DCX^+^ cells of the groups subjected to TBI (*p* < 0.029, Appendix A). At 30 days post TBI, we observe a strong increase in both hemispheres in the number of neuroblasts in the TBI groups, compared to the corresponding SHAM groups (DCX^+^ cells, ipsilateral: *p* < 0.001, Figure 3H and Figure 4A,B, Appendix A; contra-lateral: *p* < 0.001, Appendix A).

Collectively, these data highlight how the neuroblast population tends to respond late to TBI, through an increase of the cell number in the ipsilateral SVZ at 30-days following the trauma, compared to the SHAM groups.

### 2.6. Influence of p21 Deletion and Running Session on TBI-Induced Neuroblast Modulation

The analysis between the injured groups at 7 and 14 days post TBI does not show any TBI-dependent modulation of DCX^+^ cells in either of the ipsi- and the contralateral region. At 30 days after the trauma, on the other hand, we find an increase in neuroblasts in the ipsilateral SVZ of the KO RUN TBI group, compared to the other groups (KO RUN TBI vs. WT TBI = 0.015, vs. WT RUN TBI and KO TBI < 0.001, a, b, c, respectively, Figure 3H and Figure 4B,C, Appendix A).

### 2.7. TBI Triggers a Boost in Migration toward the Peri-Lesion Cortical Region of KO RUN Mice

The next step was to study the migratory flow of new cells that originated in the SVZ and were redirected to the damage site in an attempt to contribute to the tissue repair process. For this study, in the four groups subjected to trauma, both the NSPCs (Nestin GFP^+^) and neuroblasts (DCX^+^) in their path from the SVZ to the injured cortex were analyzed. We also analyzed the proliferative fraction of NSPCs and neuroblasts. The data obtained show that 7 days after TBI in the KO RUN TBI group, there is a significant increase in both NestinGFP^+^ and Ki67^+^NestinGFP^+^ cells within the migration region towards the injured cortex, compared to the other groups (NestinGFP^+^: KO RUN TBI vs. WT TBI, *p* = 0.017, vs. WT RUN TBI, *p* = 0.031 and vs. KO TBI *p* = 0.027, Figure 5A,I,J; Ki67^+^ NestinGFP^+^ cells: KO RUN TBI vs. WT TBI, *p* = 0.027, vs. WT RUN TBI, *p* = 0.038 and vs. KO TBI *p* = 0.049, Figure 5B,I,J).

In addition, 14 days after the TBI, our data show in the migratory stream (MS) of the KO RUN TBI group a significant increase, compared to the other experimental groups, in NestinGFP^+^ cells (KO RUN TBI vs. WT TBI, *p* < 0.001, vs. WT RUN TBI, *p* = 0.01 and vs. KO TBI, *p* = 0.023, Figure 5C,K,L) and DCX^+^ cells (KO RUN TBI vs. WT RUN TBI, *p* = 0.024 and vs. KO TBI, *p* = 0.02, Figure 5D,K,L). We also observed a marked enhancement in proliferating migrating cells in the KO RUN TBI group, both in terms of total proliferation (KO RUN TBI vs. WT TBI, *p* = 0.046, vs. WT RUN TBI, *p* = 0.04 and vs. KO TBI, *p* = 0.0028), and regarding the proliferative rate of Nestin GFP^+^ cells (Ki67^+^ NestinGFP^+^: KO RUN TBI vs. WT TBI, *p* = 0.007, vs. WT RUN TBI, *p* = 0.038 and vs. KO TBI, *p* = 0.0073, Figure 5E,K,L), and neuroblasts (Ki67^+^ DCX^+^ cells: KO RUN TBI vs. WT TBI, *p* = 0.004, vs. WT RUN TBI, *p* = 0.008 and vs. KO TBI, *p* = 0.0018, Figure 5F,K,L).

At 30 days post TBI, we observe a significant genotype effect with an increase in the p21 KO groups in density within the MS of Nestin GFP^+^ and DCX^+^ cells, compared to the WT animals (NestinGFP^+^: *p* = 0.011, Figure 5G; DCX^+^: *p* = 0.0015, Figure 5H).

Altogether, these data suggest that voluntary physical activity promotes widespread TBI-dependent migration from SVZ toward the injured cortex of NSCs and neuroblasts in mice lacking the p21 gene, likely enhancing the neuroprotective and regenerative processes that occur after the lesion.

### 2.8. p21 Deletion and Running Session Strongly Influence the TBI-Induced New-Born Cell Localization in the Peri-Lesion Cortex

Moreover, we characterized the presence of newborn cells, likely derived from the SVZ, in the damage site 7, 14 and 30 days after TBI. To this aim, we considered three different areas of the damage, which were as follows: (i) one area lateral to the lesion (lateral); (ii) one on the medial side (medial); (iii) the last on the central lower edge of the lesion (low). In these regions, we counted the total number of NestinGFP^+^ and DCX^+^, as well as the number of NestinGFP^+^ and DCX^+^ cells co-expressing BrdU, assuming that most of these cells that express BrdU might originate in and derive from the SVZ through the migratory redirection previously analyzed. At 7 days post TBI, we observe a substantial increase in BrdU^+^ NestinGFP^+^ and BrdU^+^ DCX^+^ cells in the KO RUN TBI group of animals, compared to the other experimental conditions both in the lateral region (BrdU^+^ Nestin GFP^+^: KO RUN TBI vs. WT TBI, *p* = 0.02, vs. WT RUN TBI, *p* = 0.041 and vs. KO TBI *p* = 0.016, Figure 6A,K,L; BrdU^+^ DCX^+^: KO RUN TBI vs. WT TBI, *p* = 0.026, vs. WT RUN TBI, *p* = 0.015 and vs. KO TBI *p* = 0.011, Figure 6B,K,L), as well as in the medial peri-lesion site (BrdU^+^ NestinGFP^+^: KO RUN TBI vs. WT TBI, *p* = 0.04, vs. WT RUN TBI, *p* = 0.027 and vs. KO TBI *p* = 0.049, Figure 6C; BrdU^+^ DCX^+^: KO RUN TBI vs. WT TBI, *p* = 0.026, vs. WT RUN TBI, *p* = 0.015 and vs. KO TBI *p* = 0.011, Figure 6D).

We also observed a genotype effect on the number of total NestinGFP^+^ and DCX^+^ cells in the lateral region (NestinGFP^+^ cells: *p* < 0.001, Appendix A; DCX^+^ cells: *p* = 0.02, Appendix A) and in the medial region (NestinGFP^+^ cells: *p* < 0.001, Appendix A; DCX^+^ cells: *p* < 0.001, Appendix A), suggesting that deletions of the p21 gene may play an important role in increasing the number of new neurons in the injured cortical regions.

At 14 days post TBI, a significant effect in the lateral peri-lesion region of p21 deletion and running on the increase in NestinGFP^+^ cells (genotype effect: *p* = 0.0013; run effect: *p* = 0.0038, Appendix A) and BrdU^+^ NestinGFP^+^ cells (genotype effect: F_(1,20)_ = 9.65, *p* = 0.0056; run effect: F_(1.20)_ = 4.64, *p* = 0.043, Appendix A) occurs. In the medial and low region of the lesion, we observed a significant increase in the DCX^+^ populations in the KO RUN TBI group (medial: KO RUN TBI vs. WT RUN TBI, *p* = 0.007 and vs. KO TBI, *p* = 0.03, Figure 6E,M,N; low: KO RUN TBI vs. WT TBI, WT RUN TBI and KO TBI, *p* < 0.001, Figure 6F) and in BrdU^+^ DCX^+^ limited to the low zone (KO RUN TBI vs. WT TBI and WT RUN TBI, *p* < 0.001, vs. KO TBI, *p* = 0.03, Figure 6G).

At 30 days post TBI, a significant effect of genotype on the increase in NestinGFP^+^ cells can be observed in the lateral peri-lesion region (lateral: genotype effect: *p* < 0.001). Moreover, we detect a significant increase in NestinGFP^+^ cells in the KO RUN TBI group when compared to the other conditions in the medial and low cerebral region (medial: KO RUN TBI vs. WT TBI, *p* < 0.001, vs. WT RUN TBI, *p* = 0.008, vs. KO TBI, *p* = 0.01, Figure 6H; low: KO RUN TBI vs. WT TBI, *p* < 0.001, vs. WT RUN TBI and KO TBI *p* = 0.01, Figure 6I,O,P). Finally, in the lateral region, we observed a strong increase in DCX^+^ cells in the KO RUN TBI animals, compared to the other three injured groups (KO RUN TBI vs. WT TBI, *p* = 0.015, vs. WT RUN TBI, and KO TBI, *p* = 0.047, Figure 6J,O,P), as well as in BrdU^+^ neuroblasts (DCX^+^ BrdU^+^ cells: KO RUN TBI vs. WT TBI, *p* = 0.013, vs. WT RUN TBI, *p* = 0.047 and vs. KO TBI *p* = 0.05).

Taken together, these data demonstrate how TBI produces a significant migratory flow of stem cells and neuroblasts towards the injured region in the KO RUN TBI group, with the consequent accumulation of these populations in the cortical area closely adjacent to the lesion.

### 2.9. Volume of the Lesion Is Not Affected by p21 Deletion or Running

Subsequently, we wanted to evaluate whether the cellular dynamics of proliferation, differentiation and especially migration that occur after TBI could modulate the macroscopic neuroanatomical recovery in our experimental groups. To this end, we measured the volume of the lesions in the cerebral cortex directly affected by the lesion of the four groups at 14 and 30 days post lesion, by the application of Cavalieri’s estimator of morphometric volume. Our analysis does not evidence any significant difference within the four experimental conditions (Figure 7A,B). Moreover, the data related to the lesion’s variation over time evidence the significant effect of time in contributing to lesion volume reduction (time effect: *p* < 0.05, Figure 7C), confirming that after TBI, the brain activates mechanisms that promote the processes of regeneration and anatomical recovery.

### 2.10. Running and p21 Deletion Induce Partial Post-TBI Functional Recovery

Since the controlled cortical impact (CCI) procedure was conducted in the cortical area corresponding to the primary motor cortex that controls the right forelimb, the next step was the evaluation of the putative relevance of injury-induced SVZ neurogenesis in ameliorating the functional recovery after TBI. To this aim, we used the Ladder Rung Walking test as a behavioral task to assess the skilled walking and right forelimb stepping, placing and coordination of the animals by measuring the number of mistakes in foot placement of the right forelimb during a 50 cm walk. We carried out this analysis at different time points, including 1 day before TBI (pre-TBI) to determine the functional baseline of the mice and 2, 7, 14 and 30 days after TBI. In the statistical analysis (ANOVA analysis for repeated measures), we considered the following four independent variables and their interaction: (1) the genotype of the animals (p21 WT or KO), (2) the surgical procedure (TBI or SHAM), (3) the physical activity (running or sedentary) and (4) the effect of the period of training (time). First of all, the data reveal the functional recovery over time, as demonstrated by the decrease in mistakes in the different TBI groups through the time points and by the statistical analysis of the time variable (time effect: *p* < 0.05, Figure 7D). Moreover, even if the statistical analysis of the independent variables genotype and running was not significant, the study of the interaction among genotype, physical activity and treatment (TBI or SHAM) was significant in influencing the functional outcome (genotype x surgery x running interaction, *p* < 0.05, Figure 7D). We did not find any significant differences among the SHAM groups that maintained a similar number of errors at every time point considered. Moreover, at P2, we noticed an increased number of errors in the four TBI groups if compared to SHAM animals, an effect that seems to be more prominent in WT RUN animals. This fact confirms that the surgical procedure by itself does not provoke any impairment in the functional performance of the animals and that contusion is a key factor for the higher number of mistakes committed by animals (TBI vs. SHAM, *p* < 0.001, Figure 7D). The graph even shows that at P7, the KO RUN TBI mice display better functional performances after the trauma than the other TBI groups (P7: KO RUN TBI vs. WT TBI, WT RUN TBI and KO TBI, *p* < 0.001). Moreover, we observe that only in KO RUN TBI mice, the number of errors is not significantly different compared to the SHAM groups at 7, 14 and 30 days after TBI (P7, P14, P33: *p* > 0.05 KO RUN TBI vs. all SHAM groups, Figure 7D). The area under the curve analysis confirms the TBI-dependent functional deterioration (TBI vs. SHAM, *p* < 0.001), while within the TBI groups, we observe a significant decrease in errors in the KO RUN TBI group compared to the other experimental groups (KO RUN TBI vs. WT TBI, *p* < 0.05, vs. WT RUN TBI and KO TBI, *p* < 0.001, Figure 7E). As a whole, our data suggest that the addictive effect of running and p21 deletion could be effective in promoting a partially higher functional outcome after TBI.

## 3. Discussion

Several studies have suggested that NSCs may retain or even potentiate their self-renew ability following injury, in order to produce additional neural progenitors and neuroblasts, which migrate toward the damaged tissue and contribute to the post-traumatic neuro-regenerative processes [32,33]. If this hypothesis is confirmed, new strategies to enhance neurogenesis could be very useful to increase the number of new neurons that can benefit the cortical region that is directly involved in the damage. In this study, we have demonstrated that the concomitant deletion of p21 and physical activity play a powerful role in enhancing the subventricular neurogenic post-traumatic response and improving functional recovery.

From the comparison between the TBI and SHAM groups, we can observe a clear dynamic in the post-traumatic SVZ neurogenic response, which results in the early activation of type B cells, followed over time by a net decrease in the Nestin GFP^+^ population and a concomitant increase in type A neuroblasts, leading to a significant increase in this population at 30 days post TBI. From such evidence, we can hypothesize that in our experimental model, TBI triggers a highly specific pro-neurogenic process within the SVZ, characterized by temporally different neurogenic responses within the sub-populations considered. These data are in apparent contradiction with a study that demonstrates that transit-amplifying cells (type C cells) are the main cell type responsible for the injury-induced increase in cell proliferation, with no contribution from either GFAP+ or DCX+ cells [34]. However, in a recent work, a gradual increase over time in proliferation (from day 1 to day 7 post TBI) associated with an increase in NestinGFP^+^/GFAP^+^ NSCs and DCX progenitors within the SVZ of rats has been observed [21]. In another single-cell transcriptomics study, it has been demonstrated that brain injury is able to transform dormant NSCs into primed quiescent and active NSCs, with the concomitant activation of protein synthesis and cell cycle genes [22]. From these and other studies clearly emerges a strong discrepancy in the post-traumatic SVZ neurogenic response, derived from the high heterogeneity of the experimental protocols used, differing significantly from each other in the type of damage applied, in the proliferative/differentiative markers used and in the length of post-traumatic time-points, as well as by the utilization of different mice strains [35].

The comparison of the post-traumatic neurogenic response within TBI groups clearly demonstrates that the p21 deletion and pre-traumatic voluntary physical activity over a 12-day period exert a potent proneurogenic effect, which belatedly results in a significant increase in the population of DCX^+^ neuroblasts at 30 days post TBI. This stimulating effect of post-TBI neurogenesis observed in the KO RUN TBI mice differs from what was observed in the KO RUN SHAM group, in which we detected a considerable increase in subventricular proliferation, which was significantly higher than in the KO RUN TBI group 7 and 14 days after injury. In this regard, we might speculate that the strong proliferative increase in the KO RUN mice at 7 and 14 days post SHAM could be the result of the hyper-proliferation of running-activated NSCs as previously observed [28], an event that was not observed in the KO RUN TBI group, which displayed a late pro-neurogenic response. These differences show how the modulation of the sub-populations of newborn cells within the SVZ is very diversified, depending on the external stimuli, and furthermore lead us to hypothesize that in the p21 ko mice, the temporally consequential effects of physical activity and trauma establish a well-defined series of cellular and microenvironment modifications that, on the one hand, require well-defined timing to optimize their proneurogenic effect whereas on the other hand, the differ profoundly from the effects observed if these external stimuli are provided separately. Alternatively, we can hypothesize that a third pro-neurogenic stimulation, the TBI, to the SVZ neurogenic niche of the KO RUN SHAM mice could lead to an excessive overload in the proliferative rate, with the consequent depletion of the NSC/progenitor pool, as previously observed following a prolonged running period (21 days) in p21 ko mice [28]. Furthermore, our study examined the effects of p21 deletion and running on the migration of newborn neurons after cortical brain injury. In the non-injured SVZ, post-mitotic neuroblasts move from SVZ to the olfactory bulb along the rostral migratory stream (RMS) [36]. When cortical regions are injured, neural progenitors start to migrate outside the RMS into the adjacent tissues, including the corpus callosum (CC) and the peri-lesioned cortex [35,37]. In this context, interesting evidence that arose from our study is represented by the observation of the widespread migration of NestinGFP^+^ and DCX^+^ cells from the SVZ towards the damaged peri-cortical regions in the KO RUN TBI group. A very intriguing aspect to take into consideration is the high fraction of proliferation observed in the migrating cells, testifying the presence of cytogenesis within the migratory flow, far from the SVZ neurogenic niche. The cytogenesis observed within the migratory flow both in the NestinGFP^+^, and to a lesser extent in the DCX^+^ cells, could represent an additional neurogenic response that is able to further increase the pool of NSCs and progenitor cells involved in the unknown tissue repair processes and post-traumatic functional improvement. Both processes are observed in all the groups subjected to TBI, although the deletion of p21 associated with physical activity greatly increases its extent. This evidence confirms what was observed in a previous study, indicating a dramatic increase in the migration of post-mitotic neuroblasts along the RMS in p21 ko mice after a 5- and 12-day running session [28]. Moreover, the widespread migration of NestinGFP^+^ cells from the SVZ towards the injured cortex could account for the transient decrease in the NestinGFP^+^ population observed within the SVZ at 14 days post TBI; in this case, a process of cellular evasion could be outlined in the phases immediately following the trauma, which leads to the gradual impoverishment of the NestinGFP^+^ population within the SVZ. An explanation for the increased cell migration in the KO RUN TBI mice is still required and further studies will evaluate the impact of p21 deletion and running on the main pathways that regulate neuroblast migration. In this regard, several groups have demonstrated the involvement of trophic factors and their receptors in the microenvironment that promotes neuroblast migration in both the naïve and post-lesioned brain, including stromal cell-derived factor 1 (SDF)/C-X-C motif chemokine 4 (CXCR4), brain-derived neurotrophic factor (BDNF)/tropomyiosin receptor kinase B (TrkB) and vascular endothelial growth factors (VEGF)/VEGF receptor [37,38,39]. In particular, it was found that BDNF caused SVZ cells to emigrate toward cerebral regions [40,41] in a concentrated manner [42]; and also that either pre- or post-traumatic exercise increased cerebral BDNF protein expression as compared to non-exercised animals [43,44,45,46], suggesting a putative role of BDNF in the increase in migration in pre-exercised animal trauma.

In our study, the large migratory flow of NSCs/neuroblasts observed in the KO RUN TBI group translates into a significant accumulation of newly generated neurons in the tissue bordering the cortical lesion. Using a long-term BrdU assay, we have also demonstrated that a large fraction of these neuroblasts are newly born and are most likely of subventricular origin. We do not know exactly the differentiative fate of these cells nor their functional role within the lesioned microenvironment, because migrating NSPCs take 1 to 3 months to fully maturate into neurons [47]; furthermore, in our study, we detected only a minimal fraction of cells that co-expressed the markers BrdU and NeuN (marker of mature neurons) at 30 days post injury, in full agreement with previous works [48,49], demonstrating the nearby absence of newly born mature neurons in peri-lesion regions. The formation of newly generated mature neurons in the cortex following injury is controversial. Recent works suggest that the cortex microenvironment may favor glial differentiation, resulting in the downregulation of DCX expression and the concomitant cortical up-regulation of Olig2 [37,50] and of Shh [51], which trigger glial differentiation [35]. Pous et al. identified the fibrinogen released in the microenvironment SVZ by the leaky vasculature after injury, a key factor driving the differentiation of NSPCs into astrocytes via the activation of the BMP signaling pathway [52]. Whatever the fate of the NSPCs of p21 runner mice after TBI, we hypothesize that their considerable increase in the peri-lesioned cortical region could be beneficial for the neuro-reparative response, as evidenced by the improvement of the Ladder Walking Test observed in the KO RUN TBI mice. In agreement with our hypothesis, Dixon et al. demonstrated that selective NSPC ablation induces a reduction in the number of neuroblasts migrating toward the injury with the consequent decrease in residential neuron and glial cells in the peri-lesion cortex and reduced locomotor recovery [53]. We believe that a strong increase in the number of NSPCs in the peri-injury cortical region of KO RUN TBI mice could greatly enhance the tissue stabilization processes of the injury milieu, allowing neuroprotection through the increased influx of neuro-protective factors, such as BDNF and VEGF, which can in turn promote neuronal survival, local glial proliferation, reduced gliosis and functional recovery. Our study is descriptive and one of the main limitations of this study is the lack of analysis of molecular mechanisms that can partially explain the results obtained. In this regard, however, some of our preliminary evaluations have highlighted the specific role of p21 deletion and physical activity in the subventricular neurogenic niche capable of downregulating the expression of anti-neurogenic genes in the BMP2 pathway. This event could trigger NSCs to exit from the quiescent state, allowing them to differentiate into the neuronal lineage [54].

## 4. Material and Methods

### 4.1. Animals

Male wild-type and p21-null mice [55] of the same genetic background (129Sv/c57BL6; 50:50; https://www.jax.org/strain/003263, accessed on 12 February 2015) were housed under a continuous 12 h light/12 h dark cycle at a constant temperature of 21 °C, with complete availability of water and food. Nestin green fluorescent protein mice (C57BL/6 background; kindly provided by Dr. G. Enikolopov) express GFP driven by the Nestin promoter [56]. Nestin-GFP mice were crossed with WT and knockout mice to obtain WT and KO/NestinGFP^+^ mice, which were interbred at least four times before further analysis, generating the different genotypes under study.

All experiments were performed blind for the different experimental conditions. We analyzed 5 mice per group in the immunohistochemistry study and 8 mice group in the functional analysis.

### 4.2. Running Paradigm and BrdU Administration

Mice subjected to the 12-day running session were housed in a running cage (2 mice per cage) and their running activity was measured with a speedometer. Moreover, the mice had been treated with BrdU administered in their drinking water (B5002, Sig-ma; 0.5 g/L) from day 7 until the end of the running session, in order to label proliferating NPCs and neuroblasts and to follow the fate of their progeny. Depending on their genotype (WT or p21 KO) and on their surgical procedure (SHAM or TBI), the mice were subdivided according to the following sedentary or running protocols: WT sedentary (WT SHAM), WT running (WT RUN SHAM), p21 ko sedentary (KO SHAM) and ko running (KO RUN SHAM), and WT sedentary (WT TBI), WT running (WT RUN TBI), p21 ko sedentary (KO TBI) and p21 ko running (KO RUN TBI).

### 4.3. Controlled Cortical Impact (CCI) Injury

Mice were anesthetized with isoflurane and positioned within a mouse stereotaxic frame. Following a longitudinal skin incision, a 3 mm diameter craniotomy was performed at the following stereotaxic coordinates: antero-posterior (AP): +0.5 mm; lateral −0.5 mm [57]. Traumatic brain injury was performed at the cortical level with a flat, 3 mm diameter metal tip attached to the CCI device (PinPoint Precision Impactor, Stoelting, Wood Dale, IL, USA), at an impact speed of 3 m/sec, time of impact of 150 ms and a depth of 2 mm below the dura, corresponding to the cerebral region of the primary cortex, which controls the fine movements of the right forelimb (TBI groups). After the impact, the animals were sutured with absorbable suture thread, housed in their home cage and put on a heated plate for 3/4 h in order to control their body temperature during their recovery from anesthesia. Animals were treated following the Italian Ministry of Health and directive 2010/63/EU guideline nr 785/19 PR. Animals subjected to the surgical procedures described above without cortical impact represented the SHAM groups.

### 4.4. Experimental Procedures

First, 12–14-week-old male p21 wt and knockout mice ran for 12 days in free running wheels; from 7 to 12 days of running, each group of animals received BrdU dissolved in drinking water, to mark the cells that underwent DNA replication and to follow the fate of their progeny. On the twelfth day of running, the mice were subjected to the controlled cortical impact (CCI) surgical procedure as described before.

The mice were sacrificed at the following three different time points: seven (P7), fourteen (P14) and thirty (P30) days after the CCI procedure (Figure 1A). By immunohistochemical assays, we analyzed the following three different regions of the injured brain: (i) the ipsi- and contralateral SVZ to the lesion to evaluate the post-traumatic neurogenic response; (ii) the migratory route of new-born cells originating in the SVZ that were re-directed toward the injured cortical regions; (iii) 3 different cortical sites (medial, low and lateral) of the lesion to detect the new-born progenitors that reached the injured area and that are supposed to contribute to ameliorating the outcome after TBI (Figure 1B). To evaluate the possible functional recovery of the mice subjected to TBI, we performed the Ladder Rung Walking Test at 2, 7, 14 and 30 days post TBI (Figure 1A).

### 4.5. Immunohistochemistry

At 7, 14 and 30 days post TBI, the animals were sacrificed by trans-cardiac perfusion with 4% paraformaldehyde (PFA) in phosphate-buffered saline (PBS); the brains were collected and kept overnight at −4 °C in PFA. They were subsequently equilibrated in sucrose diluted at 30% and finally cryopreserved at −80 °C. Slicing was carried out by embedding the brain in Tissue-Tek OCT (Sakura, Torrence, CA, USA) and then cut using a cryostat at −25 °C throughout the whole rostro-caudal extent. The coronal sections were processed in a one-in-six series protocol at a 40 μm thickness. Sections were then stained for multiple labelling using different fluorescence techniques. Sections were initially washed with 0.1 M glycine for 10 min, followed by permeabilization using 0.3% Triton X-100 in PBS for another 10 min. The sections were then incubated for 30 min in a blocking solution that contained 3% normal donkey serum (NDS) in 0.3% Triton X-100 in PBS to saturate the specific sites, followed by incubation with the same blocking solution that contained primary antibodies for 16–18 h at 4 °C. The primary antibodies used were goat polyclonal antibodies, which were used against DCX (Santa Cruz Biotechnology, Dallas, TX, USA; Cat# Sc-8066; 1:300); a rabbit monoclonal antibody was used against Ki67 (Lab Vision, South San Francisco, CA, USA, Cat# RM-9106-S; 1:150), whereas a mouse monoclonal antibody was used against GFAP (Sigma, St. Louis, MO, USA, Cat# G6171; 1:500). The detection of BrdU-positive cells consisted of denaturing DNA with 2N HCl for 45 min at 37 °C to facilitate antibody access. The sections were then incubated with 0.1 M sodium borate buffer at pH 8.5, followed by overnight incubation at 4 °C with a rat anti-BrdU primary antibody (Abcam, Cambridge, UK, Cat# ab6326; 1:300) diluted in TBS that contained 0.1% Triton, 0.1% Tween, and 3% normal donkey serum (blocking solution). To observe primary antibody binding, donkey secondary antibodies against rat (BrdU) and rabbit (Ki67) conjugated to Cy3 (Jackson ImmunoResearch, West Grove, PA, USA; 1:200 in PBS), and against goat (DCX) and mouse (GFAP) antibodies conjugated to Alexa-647 (Invitrogen, San Diego, CA, USA; 1:300 in PBS) were used. Nuclei were observed by incubating sections with Hoechst (1:500).

### 4.6. Cell Counting

Cell numbers in the SVZ were obtained with stereological analysis, by counting the cells that expressed the indicated markers and were visualized with confocal microscopy throughout the whole rostro-caudal extent of the SVZ in a one-in-ten series of 40 μm free-floating serial coronal sections (240 μm apart). The cell numbers obtained for each SVZ section were divided for the corresponding area of the section to obtain the average number of SVZ cells per 100 mm^2^. The areas were obtained by tracing the outline of the whole SVZ bulb, identified by the presence of cell nuclei stained by Hoechst on a digital picture captured and measured using ImageJ software (Version 1.52t released 30 January 2020) [58]. A CellSens Standard system (OLYMPUS) was used to record z-stack images, and thus confirm the colocalization of multiple labeled cells in the SVZ. To assess the neural stem/progenitor cell and neuroblast cell numbers in the migratory stream and perilesional region, non-biased cell number estimations were performed on the 5 most central rostro-caudal sections around the injury epicenter (as determined using cresyl violet-stained sections), which were 30 μm thick and 180 µm apart. The count of the labelled cells in the regions bordering the lesion was carried out within a frame of 300,000 mm^2^.

### 4.7. Estimation of Lesion Volume

On a Polysine microscope slide (Thermo Scientific, Waltham, MA, USA), a number of brain slices that permitted us to comprehend the entire damage extension were placed. Then, images of the sections were acquired by fluorescent microscopy at a magnification of 4×. The areas of the damage (mm^2^) on each section were estimated via the “Polygon Selection” tool of ImageJ and following this the total volume of brain damage (mm) was calculated by Cavalieri’s estimator of morphometric volume, which is as follows: VC = d (Σ yi) − (t) Ymax, where d is the distance between the sections contained in a well (d = 240 μm), yi is the area of a single section, t is the section thickness (t = 40 μm) and yMAX is the maximum value of y. The factor (t) yMAX is subtracted from the basic equation as a correction for overprojection.

### 4.8. Ladder Rung Walking Task

The apparatus is made by two see-through walls of Plexiglass of 1 m of length and 20 cm of height with removable metal rungs (3 mm of diameter) at a minimum distance of 1 cm. The ladder is placed at a minimum of 50 cm above the ground with a neutral cage on one side, from which the mice start the task, and the home cage with the littermates at the other. The width of the apparatus can be adjusted in order to prevent the animals from turning around. Animals were tested with a regularly 2 cm interspaced rung pattern one day before the CCI procedure (pre TBI) to determine the baseline scores and 2 (P2), 7 (P7), 14 (P14) and 30 (P30) days after the surgery to evaluate the functional recovery. Every mouse underwent 4 trials with a ladder length of 50 cm during which they were recorded with a camera placed at one side of the apparatus in order to obtain a clear view of the considered limb (right forelimb). The scoring was carried out through the observation of the recordings in slow-motion and counting the number of errors for each trial (foot placement accuracy analysis). The following errors were considered: total miss (the limb completely missed a rung), deep slip (the limb initially reached the rung but then slipped off, causing a fall when weight-bearing) and slight slip (the limb slipped off when weight-bearing but not causing a fall or an interruption of the gait). The mean number of errors per trial was then standardized for a ladder length of 1 m. The areas under the curve were calculated with Prism 5.

### 4.9. Statistical Analysis

The data on the cellular responses after TBI in the SVZ have been analyzed through a three-way ANOVA, with genotype, running and treatment as the independent variables. Cell migration and number of cells in the peri-lesioned area have been analyzed through a two-way ANOVA, with genotype and running as the independent variables. Lesion volumes at different time-points have been analyzed through a mixed ANOVA, with genotype, running and time as the independent variables. Finally, the behavioral data from the Ladder Rung Walking task have been analyzed through a mixed ANOVA, with genotype, running, time and treatment as the independent variables. Fisher’s LSD post hoc tests have been conducted whenever necessary. Analyses have been performed with GraphPad Prism 5 software for the two-way ANOVA and with Statistica software 14.0 (Dell Software) for the three-way and mixed ANOVA studies.

## 5. Conclusions

The data obtained in this study reveal how the interaction between physical activity and p21 gene deficiency plays an important role in neurogenic and migratory mechanisms in response to traumatic injury. Our data do not reveal whether the increase in neuroblasts within the SVZ and their subsequent migration towards the lesioned cortex is a process capable of increasing the rate of functionally active newly mature neurons. However, the data obtained in the Ladder Rung Walking Test represented an indirect clue relating the correlation within the KO RUN TBI group between the cellular processes that take place in the post-traumatic sequelae and an improvement in the functional response underlying that particular task. We believe that this study can offer interesting perspectives for future pre-clinical strategies aimed at investigating the role of physical activity and NSCs in the post-traumatic neuro-regenerative response.

## Figures and Tables

**Figure 1 ijms-24-02911-f001:**
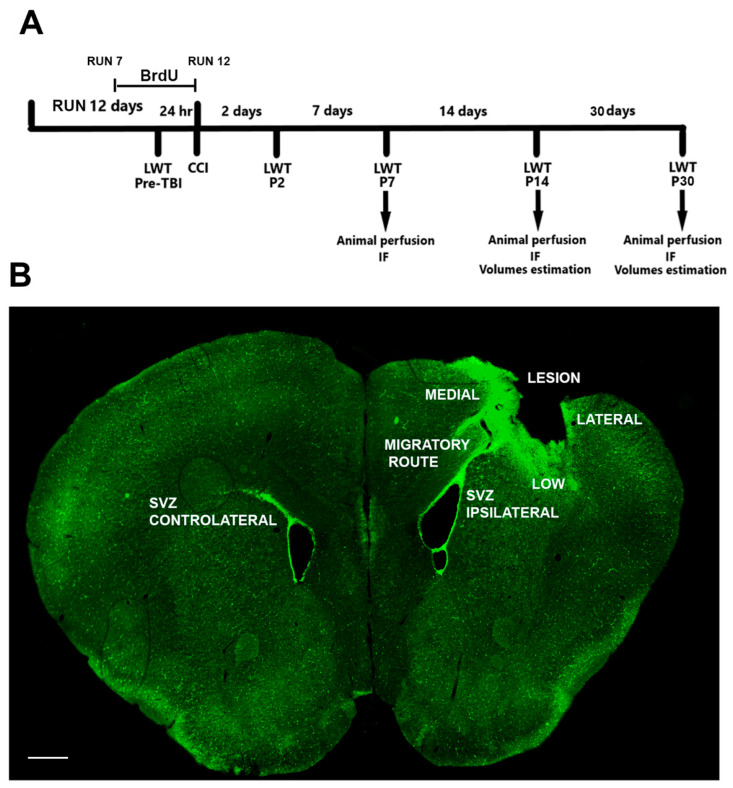
Graphical representation of experimental procedures. (**A**) Experimental timeline. The animals have been subjected to a running session for 12 days. From day 7 of the running session until the day of the lesion, BrdU is administered in the drinking water of the animals. On the 12th day, the mice undergo the CCI surgical procedure. The functional outcome has been evaluated by the Ladder Rung Walking Task (LWT) 24 h before CCI (pre-TBI) to determine the baseline number of errors, and two, seven, fourteen and thirty-three days after the surgery (P2, P7, P14 and P30). At P7, P14 and P30, the animals have been sacrificed to perform immunofluorescence assays (IF). (**B**) Coronal section of a lesioned brain. To evaluate the cellular processes that occur in the brain after the trauma, we considered the following: the ipsi- and contralateral SVZ, the migratory route and the low, medial and lateral sides of the lesioned cortex. Scale bar = 400 μm.

**Figure 2 ijms-24-02911-f002:**
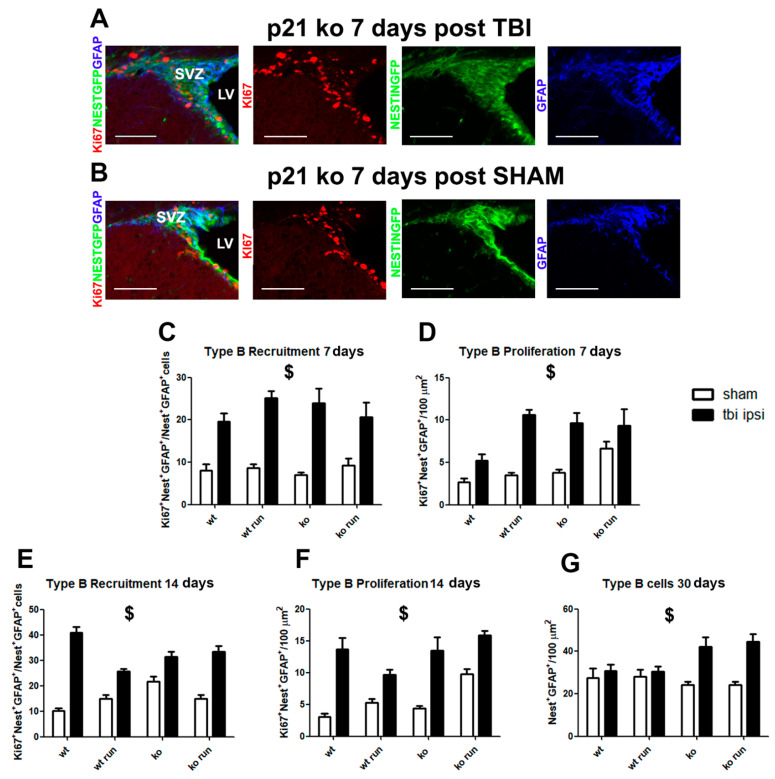
Modulation of type B cells after TBI. (**A**,**B**) Representative images in coronal sections of the neurogenic processes that occur in the ipsilateral SVZ of p21 ko mice 7 days after TBI (KO TBI, (**A**)) and after SHAM (KO SHAM, (**B**)). The images show an increment in the pool of proliferating type B cells (Ki67^+^/GFAP^+^/NestinGFP^+^ cells) in the KO TBI animals, with respect to the KO SHAM mice at 7 days after the surgery (N = 5 mice/group). (**C**) Graph shows the increase in type B recruitment at 7 days post TBI in the ipsilateral SVZ of the mice subjected to injury in comparison to their SHAM groups (ratio of Ki67^+^ NestinGFP^+^ GFAP^+^ cells/NestinGFP^+^ GFAP^+^ total cells, ipsilateral: lesion effect: F_(1,60)_ = 91.45, *p* < 0.001). (**D**) Graph shows the enhancement of type B proliferation in the ipsi-lateral SVZ of TBI groups (Ki67^+^ NestinGFP^+^ GFAP^+^ cells, ipsilateral: lesion effect: F_(1,60)_ = 45.8 *p* < 0.001). (**E**) Histograms illustrate the increase in type B recruitment in the ipsilateral SVZ in mice after 14 days from TBI (lesion effect: F_(1,56)_ = 188, *p* < 0.001, $). (**F**) Graphs indicate an increase in type B proliferation in the ipsi-lateral SVZ of mice subjected to TBI 14 days after the trauma (lesion effect: F_(1,56)_ = 85.9, *p* < 0.001). (**G**) After 30 days from TBI, we observed a significant increase in the type B cell population in the TBI mice with respect to their SHAM groups (NestinGFP^+^ GFAP^+^ cells, ipsilateral: lesion effect: F_(1,44)_ = 21.76, *p* < 0.001, $). Statistical significance of main lesion effect between SHAM and TBI groups: $ *p* < 0.001. Multifactorial analysis with the following three independent variables: genotype, treatment and running, followed by Fisher’s LSD post hoc tests. Magnification = 20×. Scale bar = 100 μm. SVZ = subventricular zone. LV = lateral ventricle.

**Figure 3 ijms-24-02911-f003:**
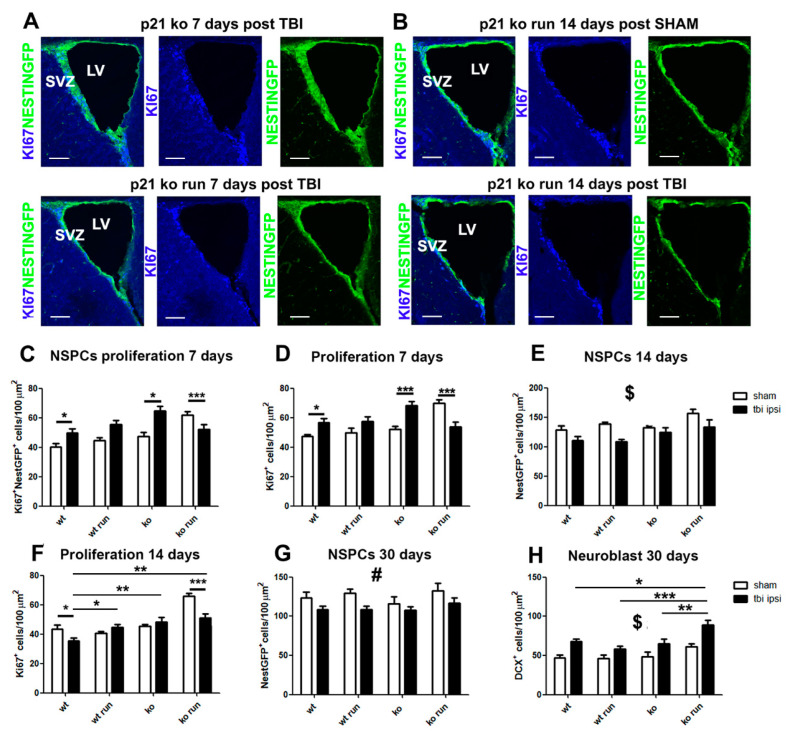
Time course of NSPCs and type A neuroblast following TBI. (**A**) Micrographs show the decreased proliferation of neural stem/progenitor cells (NSPCs, Ki67^+^/NestinGFP^+^ cells) in the KO TBI RUN group with respect to the KO TBI mice at 7 days post TBI. (**B**) Representative pictures indicate the significantly decreased number of Ki67^+^/NestinGFP^+^ cells in the KO TBI RUN mice in comparison to their respective SHAM littermate 14 days after TBI. (**C**) Histograms show a significant increase in the proliferating NSPCs of WT TBI and KO TBI mice in comparison with their respective WT SHAM and KO groups (Ki67^+^ NestinGFP^+^ cells, ipsilateral: genotype x run x lesion interaction: F_(1,112)_ = 11.71, *p* < 0.001, followed by LSD post-test, WT TBI vs. WT SHAM and KO TBI vs. KO SHAM *p* = 0.015). (**D**) Graph indicates a significant increase in the total proliferation of WT TBI and KO TBI mice with respect to the WT and KO SHAM groups (Ki67^+^ cells, genotype x run x lesion interaction, ipsilateral: F_(1,108)_ = 14.38, *p* < 0.001, followed by LSD post-test, WT TBI vs. WT SHAM, *p* = 0.03, KO TBI vs. KO SHAM *p* < 0.001). (**C**,**D**) It is also possible to observe the decrease in the NSPCs proliferation of KO RUN TBI mice with respect to the KO RUN SHAM group (KO RUN TBI vs. KO RUN SHAM, *p* < 0.001, (**C**)) and total proliferation (KO RUN TBI vs. KO RUN SHAM, *p* < 0.001, (**D**)). (**E**) Graph displays the reduced proliferation of NSPCs in mice analyzed 14 days after TBI, when compared to their respective SHAM counterparts (NestinGFP^+^ cells, ipsilateral: lesion effect: F_(1,105)_ = 17.21, *p* < 0.001, $). (**F**) Histogram shows the decreased total proliferation after 14 days after TBI in the WT TBI and KO RUN TBI mice with respect to their SHAM littermates (Ki67^+^ cells: genotype x run x lesion interaction: F_(1,99)_ = 17.04, *p* < 0.001, followed by LSD post-test, WT TBI vs. WT SHAM, *p* = 0.029; KO RUN TBI vs. KO RUN SHAM *p* < 0.001). In the TBI groups, we detected a significant increase in proliferation in the WT RUN TBI mice (F_(1,108)_ = 5.84, *p* = 0.017, followed by LSD post-test, WT TBI vs. WT RUN TBI, *p* = 0.01), as well as an increase in the KO TBI and KO RUN TBI mice (*p* < 0.01), in comparison with the WT TBI group. (**G**) Graph shows the decreased NSPC population in the ipsilateral SVZ of TBI mice with respect to their SHAM genotypes, 30 days after the TBI (NestinGFP^+^ cells: lesion effect: F_(1,89)_ = 9.2, *p* = 0.003, #). (**H**) Histogram indicates at 30 days post TBI the increased number of neuroblasts (DCX^+^ cells) in the TBI group in comparison with their SHAM littermates (DCX^+^ cells, ipsilateral, lesion effect: F_(1,53)_ = 20.7, *p* < 0.001, $). The comparison within the TBI groups (black histograms) show a significant increase in neuroblasts in the KO RUN TBI mice in comparison with WT TBI, WT RUN and KO TBI mice (genotype x run interaction: F_(1,53)_ = 7.77, *p* = 0.007, followed by LSD post-test, KO RUN TBI vs. WT TBI = 0.015, vs. WT RUN TBI < 0.001, vs. KO TBI = 0.001, a, b, c, respectively). N = 5 mice/group. Statistical significance of LSD post hoc analysis: * *p* < 0.05, ** *p* < 0.01 and *** *p* < 0.001. Statistical significance of main lesion effect between SHAM and TBI groups: $ *p* < 0.001 and # *p* < 0.01. Multifactorial analysis with the following three independent variables: genotype, treatment and running, followed by Fisher’s LSD post hoc tests. Magnification = 20×. Scale bar = 100 μm. SVZ = subventricular zone. LV = lateral ventricle.

**Figure 4 ijms-24-02911-f004:**
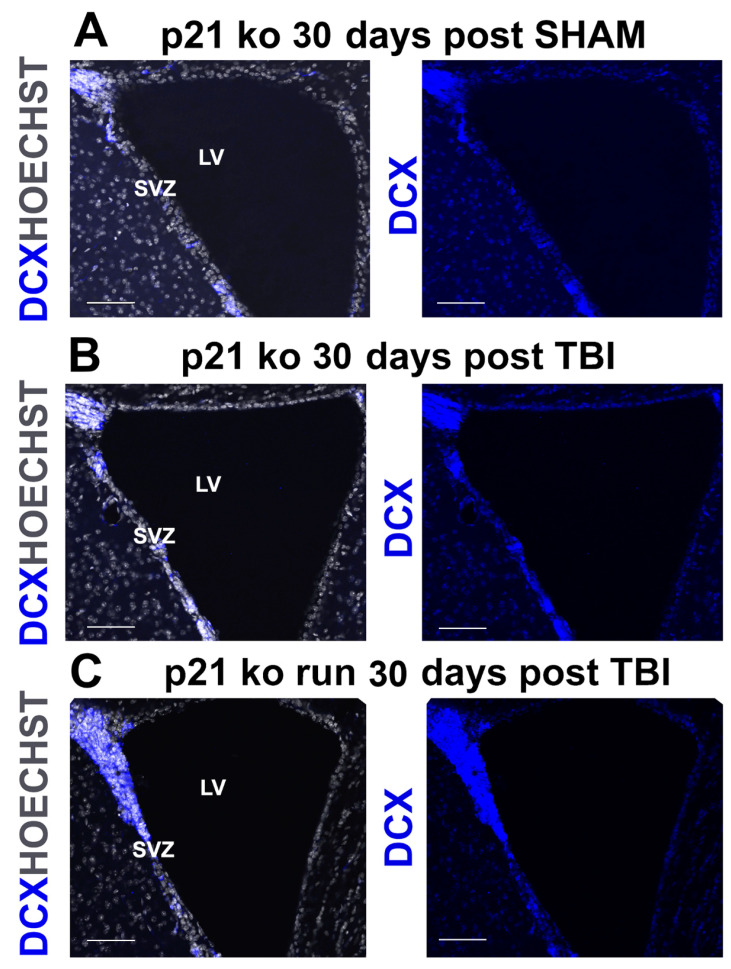
(**A**–**C**). Increase in DCX+ cells in the KO RUN TBI group. Confocal representative micrographs show the increased number of DCX^+^ cells in the KO RUN TBI mice (**C**), in comparison with the KO SHAM (**A**) and KO TBI (**B**) mice, 30 days after TBI. Magnification = 20×. Scale bar = 100 μm. SVZ = subventricular zone. LV = lateral ventricle.

**Figure 5 ijms-24-02911-f005:**
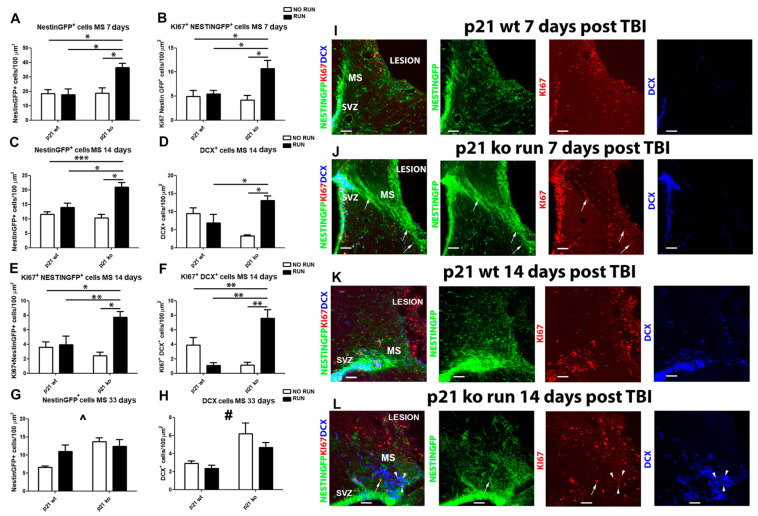
Cell migration following TBI. (**A**) Graph shows the increase in the migratory stream (MS) of the KO RUN TBI mice in comparison to the other groups 7 days after the lesion of NestinGFP^+^ cells (genotype x running: F(_1.18_) = 5.98, *p* = 0.025, followed by LSD post-test, KO RUN TBI vs. WT TBI, *p* = 0.017, vs. WT RUN TBI, *p* = 0.031 and vs. KO TBI *p* = 0.027). (**B**) Graph shows the increase in proliferating NSPCs in the migratory stream (MS) of the KO RUN TBI mice 7 days after the lesion (Ki67^+^ Nestin GFP^+^ cells: genotype x running: F(_1,18_) = 5.51, *p* = 0.03, followed by LSD post-test, KO RUN TBI vs. WT TBI, *p* = 0.027, vs. WT RUN TBI, *p* = 0.038 and vs. KO TBI *p* = 0.049). (**C**) Histograms illustrate the enhancement in the MS of KO RUN TBI mice with respect to the other experimental condition 14 days after the TBI of NestinGFP^+^ cells (genotype x run interaction: F_(1,18)_ = 6.55, *p* = 0.019, followed by LSD post-test, KO RUN TBI vs. WT TBI, *p* < 0.001, vs. WT RUN TBI, *p* = 0.01 and vs. KO TBI, *p* = 0.023). (**D**) Graph shows the increment, with respect to the other experimental conditions of DCX^+^ cells, in the MS of KO RUN TBI 14 days after TBI (genotype x run interaction: F_(1,18)_ = 7.07, *p* = 0.016, followed by LSD post-test, KO RUN TBI vs. WT RUN TBI, *p* = 0.024 and vs. KO TBI, *p* = 0.02). (**E**) Graph shows the increase in the MS of KO RUN TBI mice with respect to the other groups 14 days after the TBI of proliferating NestinGFP^+^ (Ki67^+^/NestinGFP^+^: genotype x run interaction: F_(1,18)_ = 5.07, *p* = 0.037, followed by LSD post-test, KO RUN TBI vs. WT TBI, *p* = 0.007, WT RUN TBI, *p* = 0.038 and vs. KO TBI, *p* = 0.0073). (**F**) Graph shows the increments, with respect to the other experimental conditions of proliferating DCX^+^ cells, in the MS of KO RUN TBI 14 days after TBI (Ki67^+^ DCX^+^ cells: genotype x run interaction: F_(1,18)_ = 7.07, *p* = 0.016, followed by LSD post-test, KO RUN TBI vs. WT TBI, *p* = 0.004, WT RUN TBI, *p* = 0.008 and vs. KO TBI, *p* = 0.0018). (**G**) Graph indicates an increase in migrating NestinGFP^+^ cells in the KO TBI and KO TBI RUN mice with respect to the WT TBI mice 30 days after TBI (genotype effect: F_(1,25)_ = 3.34, *p* = 0.011, ^). (**H**) Histograms show an enhancement of migrating DCX^+^ cells in the KO TBI and KO TBI RUN group with respect to the WT TBI mice 30 days after TBI (genotype effect: F_(1.25)_ = 12.64, *p* = 0.0015, #). (**I**,**J**) Confocal micrographs show the increased density of migrating NestinGFP^+^ cells observed in the KO RUN TBI mice with respect to the WT TBI mice, 7 days from TBI. (**K**,**L**) Confocal micrographs illustrate the enhancement of NSPCs (NestinGFP^+^) and neuroblasts (DCX^+^) along the MS of the KO RUN TBI mice compared to the WT TBI group, 14 days post TBI. Arrow indicates the presence of proliferating NestinGFP^+^ cells and arrowheads indicate proliferating DCX^+^ cells. N = 5 mice/group. Statistical significance of LSD post hoc analysis: * *p* < 0.05, ** *p* < 0.01 and *** *p* < 0.001. Statistical significance of main genotype effect between WT and KO groups: # *p* < 0.01, ^ *p* < 0.05. Two-way ANOVA analysis followed by Fisher’s LSD post hoc tests. Magnification = 20×. Scale bar = 100 μm. SVZ = subventricular zone. MS = migratory stream.

**Figure 6 ijms-24-02911-f006:**
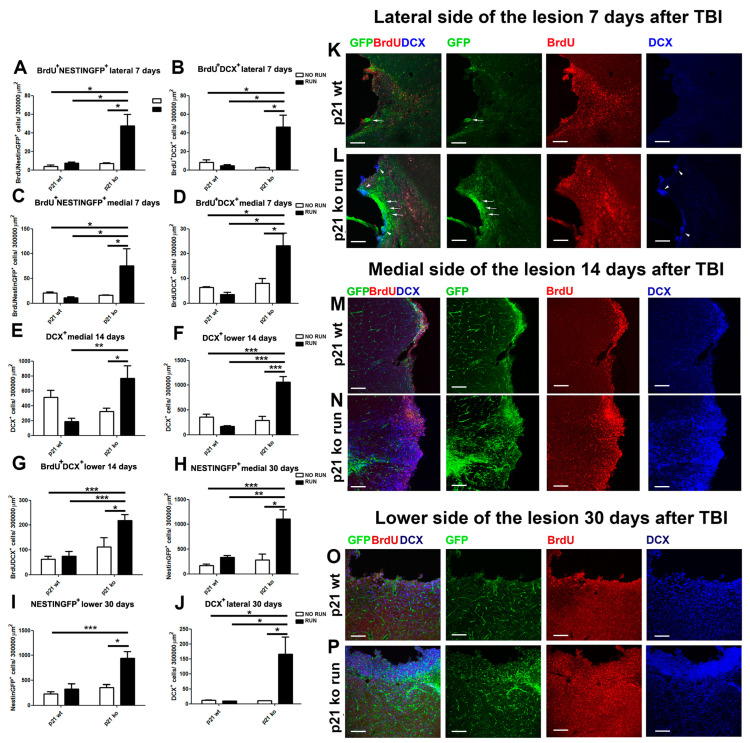
Distribution of NSPCs and neuroblasts in the peri-lesion cortex. **Seven days post TBI** (**A**,**C**). Graphs show the increased density in the lateral (**A**) and medial (**C**) regions that border the cortical lesion of the KO RUN TBI mice of Brdu^+^NestinGFP^+^ cells (lateral, genotype x running: F_(1,24)_ = 4.64, *p* = 0.041, followed by LSD post-test, KO RUN TBI vs. WT TBI, *p* = 0.02, vs. WT RUN TBI, *p* = 0.041 and vs. KO TBI *p* = 0.016, (**A**); medial, genotype X running: F_(1,21)_ = 7.26, *p* = 0.013, followed by LSD post-test, KO RUN TBI vs. WT TBI, *p* = 0.04, vs. WT RUN TBI, *p* = 0.027 and vs. KO TBI *p* = 0.049, (**C**)). (**B**,**D**) Histograms illustrate in the lateral (**B**) and medial (**D**) peri-lesioned cortical area of the KO RUN TBI mice the enhanced BrdU^+^DCX^+^ cell density (lateral, genotype X running: F_(1,21)_ = 7.32, *p* = 0.012, followed by LSD post-test, KO RUN TBI vs. WT TBI, *p* = 0.026, vs. WT RUN TBI, *p* = 0.015 and vs. KO TBI *p* = 0.011, (**B**); medial, BrdU^+^ DCX^+^: genotype X running: F_(1,21)_ = 7.32, *p* = 0.012, followed by LSD post-test, KO RUN TBI vs. WT TBI, *p* = 0.026, vs. WT RUN TBI, *p* = 0.015 and vs. KO TBI *p* = 0.011, (**D**)). **Fourteen days post TBI.** (**E**,**F**,**G**) Graphs show, in the medial (**E**) and low (**F**) peri-injured cortical region of the KO RUN TBI, a significant increase in DCX^+^ cells (medial, genotype x run interaction: F_(1,20)_ = 14.14, *p* = 0.0012, followed by LSD post-test, KO RUN TBI vs. WT RUN TBI, *p* = 0.007 and vs. KO TBI, *p* = 0.03, (**E**) and low (genotype x run interaction: F_(1,20)_ = 7.52, *p* = 0.012, followed by LSD post-test, KO RUN TBI vs. WT TBI and WT RUN TBI, *p* < 0.001, vs. KO TBI, *p* = 0.03, (**F**)), as well as an increase in BrdU^+^DCX^+^ cells in the low side ((**G**), genotype x run interaction: F_(1,20)_ = 7.52, *p* = 0.012, followed by LSD post-test, KO RUN TBI vs. WT TBI and WT RUN TBI, *p* < 0.001, vs. KO TBI, *p* = 0.03). **Thirty days post TBI.** (**H**,**I**) Histograms illustrate the significant increment in the KO RUN TBI group of NestinGFP^+^ cells in the medial (**H**) and low (**I**) regions (medial: genotype x run interaction: F_(1,15)_ = 8.26, *p* = 0.011, followed by LSD post-test, KO RUN TBI vs. WT TBI, *p* < 0.001, vs. WT RUN TBI, *p* = 0.008, vs. KO TBI, *p* = 0.01, (**H**); low: genotype x run interaction: F_(1,15)_ = 6.71, *p* = 0.02, followed by LSD post-test, KO RUN TBI vs. WT TBI, *p* < 0.001, vs. WT RUN TBI and KO TBI *p* = 0.01, (**I**)). (**J**) Graph shows the increase in DCX^+^ cells in the lateral cortical regions of KO RUN TBI mice, 30 days after TBI (genotype x run interaction: F_(1,15)_ = 6.51, *p* = 0.022, followed by LSD post-test, KO RUN TBI vs. WT TBI, *p* = 0.015, vs. WT RUN TBI, and KO TBI, *p* = 0.047). (**K**,**L**) Confocal representative pictures show the specific localization of NestinGFP^+^ (arrows) and DCX^+^ (arrowheads) cells in the lateral cortical region lining the lesion of the KO RUN TBI mice, at 7 days post TBI (**K**). At the same time-point in the WT TBI group, we observe only a limited number of NestinGFP^+^ (arrow) cells in the peri-lesioned lateral side. (**M**,**N**) Confocal representative micrographs show that after 14 days from TBI, it is possible to observe in the medial side of the lesion of KO RUN TBI mice (**M**) a high density of NestinGFP^+^ (green), and DCX^+^ (blue) cells co-localizing with BrdU (red). In the same area of the WT TBI (**N**) mice, we detect a much lower density of cells. (**O**,**P**) The confocal pictures show the accumulation of DCX^+^ and BrdU^+^DCX^+^ cells in the low cortical peri-lesion area of the KO RUN TBI mice (**P**), which is not detectable in the WT TBI group (**O**). N = 5 mice/group. Statistical significance: * *p* < 0.05, ** *p* < 0.01 and *** *p* < 0.001. Two-way ANOVA analysis and Fisher’s LSD post hoc tests. Magnification = 20×. Scale bar = 100 μm.

**Figure 7 ijms-24-02911-f007:**
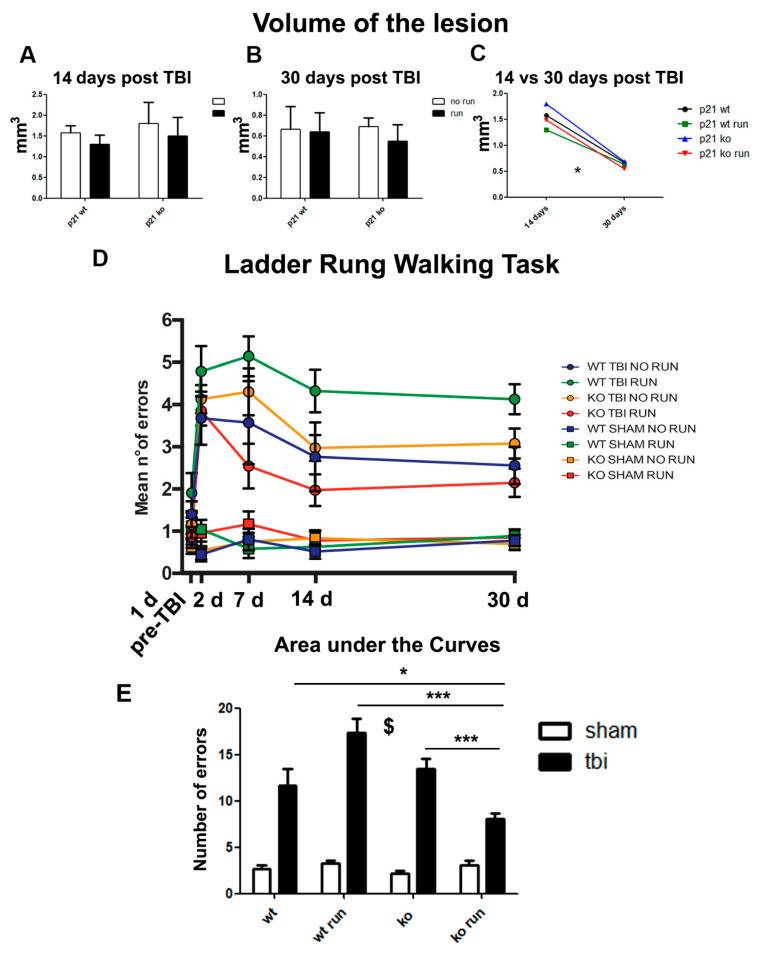
Anatomical and functional recovery following TBI. (**A**,**B**) Graphs show the average volumes of the lesions 14 (**A**) and 30 (**B**) days after the TBI procedure in the four experimental groups. (**C**) Histograms indicate the decreased average volumes of the lesions from 14 to 30 days post TBI in the four different experimental conditions (time effect: F_(4,28)_ = 22.08; *p* < 0.05). (**D**) The graph indicates the mean number of errors per group at each time point. The same color shows an experimental condition and its relative SHAM control. The statistical analysis evidenced significant effects of the time variable (time effect F = 9.949 *p* < 0.05, Figure 7D). Within the TBI groups, we found that at 7 days post TBI, the KO RUN TBI animals demonstrated better performances with respect to the other conditions and, notably, a number of mistakes comparable to their SHAM control time effect (P7: genotype x run interaction: F_(1,64)_ = 7.65 *p* = 0.032, followed by LSD post-test, KO RUN TBI vs. WT TBI, WT RUN TBI and KO TBI, *p* < 0.001). For the statistical analyses of the volumes of the lesions, we performed a multifactorial analysis with the following three independent variables: genotype; running and time. (**E**) The histograms represent the area under the curve analysis and show the increase in errors in the TBI groups (lesion effect F_(1,48)_ = 153; *p* < 0.001, $); moreover, in the TBI groups, the statistical analysis indicates a decrease in errors in the KO RUN TBI mice in comparison with the other TBI groups (genotype x run x lesion interaction, F_(1,48)_ = 12.92, *p* < 0.001, followed by LSD post-test, KO RUN TBI vs. WT TBI, *p* = 0.011, vs. WT RUN TBI, and KO TBI, *p* < 0.001). The behavioral data of the Ladder Rung Walking task have been analyzed by a multifactorial analysis with the following four independent variables: genotype, running, time and the treatment. The area under the curve statistics were evaluated by multifactorial analysis with the following three independent variables: genotype, treatment and running. (N = 8 mice/group). The post hoc analyses have been conducted via by Fisher’s LSD post hoc tests. Statistical significance: * *p* < 0.05 and *** *p* < 0.001.

## Data Availability

Not applicable.

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
