# Peer review of "Role of Running-Activated Neural Stem Cells in the Anatomical and Functional Recovery after Traumatic Brain Injury in p21 Knock-Out Mice"

_ijms, 2023, doi:10.3390/ijms24032911_

Round 1

Reviewer 1 Report

Dear Authors,

please find my suggestion to improve your manuscript.

The paper is very complex since contains a lot of data and it should be improved in some respects.

Major points

The authors described twice the animal groups

Line 455

Mice were subdivided in 4 different groups: i) p21 sedentary wt (WT), ii) p21sedentary ko (KO), iii) p21 wt running (WT RUN), iv) p21 ko running (KO RUN).

Line 492

In this study we analyzed 4 groups of SHAM mice: p21 wt sedentary (WT SHAM), p21 wt running (WT RUN SHAM), p21 ko sedentary (KO SHAM) and p21 ko running (KO RUN SHAM), and 4 groups of TBI mice: p21 wt sedentary (WT TBI), p21 wt running (WT RUN TBI), p21 ko sedentary (KO TBI) and p21 ko running (KO RUN TBI).

Maybe you should say in a complete way only one time

Line 463

I do not understand, how the size of the craniotomy is inferior to the diameter of the metal tip?

Mice were anesthetized with isoflurane and positioned within a mouse stereotaxic frame. Following a longitudinal skin incision, a 2 mm diameter craniotomy was made centered at 0.5 mm anterior to bregma and 0.5 mm lateral to the midline [56]. Cortical injury was performed with a flat, 3 mm diameter metal tip connected to the CCI device (PinPoint Precision Impactor, Stoelting), at impact speed 3m/sec, 150 ms time of impact and depth of 2 mm below the dura, corresponding to the cerebral region of the primary cortex which controls the fine movements of right forelimb.

Line 465

(stereotaxic coordinates from Bregma: antero-posterior (AP) 0.5; lateral 0.5, 34).

Please specify stereotaxis coordinates from Bregma: AP (antero-posterior)=a mm; ML (medial-lateral)=b mm; DV (dorso-ventral=c mm. I don’t understand what 34 means, is it a reference?

FIG 2

It is not clear how, in fig 2G, wt and wt run, sham results significantly lower than the respective tbi. From the figure, it does not look alike.

FIG 3

Some other points I don’t understand, concern the figure 3C, 3D, and 3E where in some cases the comparison between the bars do not correspond to the indication of significance the authors point out.

FIG 6

The figure is typed differently in A, B, C, D, E, F, G, H, I, J, it would be advisable to standardize the points and eliminate the legend NO RUN-RUN in graph A, C, E, G, I.

FIG 7

In figure 7D, the X-axis on the graph should be linear (the space between pre-tbi - p2, p2- p7, p7- p14, p14-p30 are the same).

Supplementary Fig 1E

See the concern expressed for Fig 2G.

Supplementary Fig 2C

See the concern expressed for Fig 2G.

Minor points

Cell markers are typed differently in various parts of the text and figure legend, for example

Line 105

Ki67+GFP+GFAP+ cells

Line 112

Nestin GFP+ GFAP+ cells   

It might be helpful to standardize the formulation.

Line 173

WT TBI vs WT RUN TBI, p = 0.01 ^, vs KO TBI, and KO RUN TBI, p <0.001 §,

Symbol ^ or § are misleading. For clarity, it would be better to place them only in the legend of the figures.

Line 446

c57BL6 should be changed in C57BL/6

Fig 2B

Ki67 in the green text should be changed into the red text

Author Response

Thank you very much for your concerns aim to improve the quality of the research. I hope to have fully answered and fully clarified your doubts.

sincerely

stefano farioli vecchioli

Reviewer 2 Report

In the manuscript by Battistini et al, the authors aimed to investigate whether the combination of exercise and p21 ablation affects neurogenesis in the SVZ and cortical neurorepair in mice exposed to the TBI paradigm. They found that neurogenesis in the SVZ was similarly affected by TBI in all groups, regardless of genotype or exercise. In contrast, p21 deletion and exercise strongly enhanced migration toward the peri-lesion and localization of newborn cells in the peri-lesion cortex. This is a well-designed and well-written paper for which I have few comments.

Since there are some statistically significant differences on the contralateral side regarding the neurogenic process between the groups, it would be better to show Supplementary Figures 1 and 2 in the main article.

Some of the results described in the results section are not shown. It would be better to show these results in the supplemental figures as well.

It is not always clear in the diagrams whether the statistical differences relate to the post-hoc analysis or simply to the main effects (exercise, genotype, or treatment).

For the ladder rung walking task, it might be interesting to calculate the area under the curve and analyze whether the differences between groups are statistically significant

In the statistical analysis, the authors state that All analyzes have been performed with GraphPad Prism 5 software. However, the GraphPad Prism 5 version does not offer the possibility to perform a three-way ANOVA.

In Figure 2B, KI67 should be shown in red.

Author Response

(The authors gave the same response as above.)

Reviewer 3 Report

Authors have shown that physical exercise (voluntary running session) induces an enhancement of adult neurogenesis in a mouse model with the deletion of p21. During the following study Authors investigated the effect of physical load stimulus on subventricular (SVZ) neurogenesis and neurorepair processes in the same mouse model (p21 knock-out mice) that undergoes to traumatic brain injury (TBI). The problem of TBI-induced neurological damage is quite important as there are no available effective treatments nowadays.

Using model organism in the study allows to demonstrate mechanisms of post-trauma neuro-regenerative processes which is quite interesting as it unveils the influence of the running before the TBI on the neuroblasts amount in the SVZ, the migration of cells to the damaged cortical region, the process of neurons differentiation in the peri-lesioned area, and might be used to plan further experiments to study (and probably accelerate) endogenous neuro-regenerative responses following brain trauma in adults.

Complex research had been planned and conducted which includes the study of neuroanatomical and functional recovery processes following traumatic brain injury and is based on the previous experiments results which allowed to choose optimal physical activity to enhance SVZ neurogenesis in mice and decrease the amount of animals in the current experiments.

The following comments do not diminish the value of the Article:

Probably it would be better to put into the title the whole name of the disease: Traumatic Brain Injury (TBI). But it is very much upon Authors decision.

Line 67 The abbreviature ‘CKIs’ has to be described.

Line 122 The following phrase should be corrected: ‘there no changes were detectable’.

Line 133 ‘emisphere.’ – typo should be corrected.

It would be a bit easier to accept the information from the Results paragraph if the figures would be located at those places in the text where they are first mentioned.

Line 198 It would be better to locate the title ‘Influence of p21 deletion and running session on TBI-induced neuroblasts modulation’ on the next page.

Line 219 The following fragment ‘C, K, L’ should be replaced with ‘C, K, L)’.

Line 221 The following fragment ‘KO RUN TBI’ should be replaced with ‘(KO RUN TBI’.

Line 229 Is that significant to put ‘p21’ indicator near the WT abbreviature? But it is very much upon Authors opinion, probably it has to stress the specificity of the mouse model and wild type animal.

Line 235 The following text ‘in the peri-lesion cortex.’ should be replaced with ‘in the peri-lesion cortex’.

Line 293 The abbreviature ‘CCI’ should be described.

Line 304 ‘First at all’ – typo should be corrected.

Line 377 ‘]28]’typo should be corrected.

Line 380 ‘]36]’ typo should be corrected.

Line 435 ‘we’ should be replaced with ‘We’.

Line 475 ‘Experimental procedures.’ – a dot at the end of the title should be removed.

Line 579 A scale bar to the Figure 1, B should be added.

Line 678 Probably Standard Errors (or Standard Deviations) of Means should be added to Figure 7, D.

It would be clearer if all the Figures would have visible scale bars.

Graphical abstract would give a reader the clue to get the idea of the research, so it would be probably good to add it.

The paragraph ‘Conclusions’ is not mandatory, but it would help to resume the information which is provided.

It would be better to check the References which should be described according the information provided on the Journal’s website.

Author Response

(The authors gave the same response as above.)

Reviewer 4 Report

In this manuscript, the authors performed in vivo animal experiments to demonstrated that concomitant deletion of p21 and physical activity play a powerful role in promoting the subventricular neurogenic post-traumatic response and functional recovery. In general, this topic is interesting for a broad readership, dealing with the role of activated neural stem cells in anatomical and functional recovery after injury. However, there are still a lot of concerns as noted below.

1.    In the present work, the authors discovered that concomitant deletion of p21 and physical activity can increase the subventricular neurogenic post-traumatic response and improve functional recovery. However, it is only limited to in vivo function research, and lacks deep mechanism exploration. What's the relationships between running and P21 deletion? How do they influence the function of neural stem cells?

2.    In the section of Material and Methods, the authors mentioned there were 4 groups of SHAM mice and 4 groups of TBI mice. But immunofluorescence staining showed only two or four groups of results, which were inconsistent with the statistical analytic results. For example, there was 2 groups in Figure 2A and 2B, p21 ko 7 days post TBI and p21 ko 7 days post sham. In addition, immunofluorescence results at the other time points (Day 14 and Day33) were not provided. At least it should be given in the supplement.

3.    The result descriptions in this paper are tedious and complex, and it is suggested to simplify.

4. There was no caption in all the figures. Please explain the abbreviations in the figure legend and indicate the sample size for each quantification in the figure legend.

Author Response

Thank you very much for your concerns aim to improve the quality of the research. I hope to have fully answered and fully clarified your doubts.

your sincerely,

stefano farioli vecchioli 

Round 2

Reviewer 4 Report

Thanks for the response. In the revised manuscript, the authors have addressed most of my concerns. However, one point remains.

1. There was no caption in all the figures (Figure 1,2,3……). For example, in Figure 1, the authors only listed what were A and B respectively. You should give the title of Figure 1 based on the results of this section.

Author Response

Thank you for your suggestion. In the revised version of the manuscript we have added the missing captions to all the figures and a title summarizing the findings of the results section.